# Fly Ash-Added, Seawater-Mixed Pervious Concrete: Compressive Strength, Permeability, and Phosphorus Removal

**DOI:** 10.3390/ma15041407

**Published:** 2022-02-14

**Authors:** Sangchul Hwang, Jung Heum Yeon

**Affiliations:** Civil Engineering Program, Texas State University, San Marcos, TX 78666, USA

**Keywords:** mix optimization, off-spec fly ash, pervious concrete, phosphorus, seawater, sustainability

## Abstract

A mix proportion of off-spec fly ash (FA)-added, seawater-mixed pervious concrete (SMPC) was optimized for compressive strength and permeability and then the optimized SMPC was tested for the rate and extent of aqueous phosphorus removal. An optimum mix proportion was obtained to attain the percentages (% wt.) of FA-to-binder at 15.0%, nano SiO_2_ (NS)-to-FA at 3.0%, liquid-to-binder at 0.338, and water reducer-to-binder at 0.18% from which a 7-day compressive strength of 14.0 MPa and a permeability of 5.5 mm/s were predicted. A long-term maximum compressive strength was measured to be ~16 MPa for both the optimized SMPC and the control ordinary pervious concrete (Control PC). The phosphorus removal was favorable for both the optimized SMPC and the Control PC based on the dimensionless Freundlich parameter (1/n). Both the optimized SMPC and Control PC had a first-order phosphorus removal constant of ~0.03 h^−1^. The optimized SMPC had a slightly lower capacity of phosphorus removal than the Control PC based on the Freundlich constant, K_f_ (mg^1−1/n^ kg^−1^ L^1/n^): 15.72 for the optimized SMPC vs. 16.63 for Control. This study demonstrates a cleaner production and application of off-spec FA-added, seawater-mixed pervious concrete to simultaneously attain water, waste, and concrete sustainability.

## 1. Introduction

Pervious concrete pavement is one of the most effective management practices for stormwater runoff control [1]. It reduces the necessity of water detention ponds, allows groundwater recharge, and reduces pollutants to improve water quality [2]. Equally important, pervious concrete pavement provides enhanced safety while reducing ponding and hydroplaning as a transportation surface [3,4].

More than 4 billion metric tons worldwide of cement, an essential component of concrete infrastructure and building construction, are currently produced [5]. In fact, cement manufacturing is responsible for ~5% of the total anthropogenic CO_2_ emissions to the atmosphere [6,7]. Coal fly ash (FA) is an industrial by-product obtained from coal-fueled power plants and makes up one of the largest problematic industrial waste streams on the earth [8]. Improper disposal of FA disrupts ecosystems, pollutes soil and water, and causes other environmental threats [9,10,11]. To achieve sustainability in both concrete production and FA management, FA has been used as a partial replacement of Portland cement in concrete production [12,13]. In general, the addition of FA has shown an improved workability of the freshly mixed concrete although it typically reduces the rate of early-age strength development [14]. In a hardened state, FA addition generally enhances the mechanical properties and durability of concrete by lowering the amount of calcium hydroxide (Ca(OH)_2_ or CH) as FA reacts with CH to form additional calcium silicate hydrate (3CaO·2SiO_2_·4H_2_O or C-S-H) gel [15,16,17]. However, the physiochemical characteristics of FA vary, depending not only on the types of coal used in a process but also on the types of process (boiler, gas control equipment, etc.) [18]. In the US only, more than 30% of the total FA generated per year is considered as off-spec FA and is landfilled, potentially causing different environmental and health problems [19].

Water is the key ingredient for cement hydration to form hydrates to bond the concrete mix together. A water-to-cement ratio typically falls in 0.4 to 0.6 [20]. As such, global water consumption in concrete production is estimated to be more than 2 billion metric tons annually. Given the current freshwater stress and the future freshwater shortages, seawater has been a fit-for-purpose alternative mixing water for concrete production [21,22,23,24]. However, the applicability of seawater in concrete production is limited due to its high chloride content that induce corrosion of the reinforcing steel bars [25,26,27], although the corrosion potential can be reduced by using polymer-coated rebars [28]. It should be noted that pervious concrete is typically produced without reinforcing steel bars (commonly known as rebars), as opposed to ordinary concrete. Therefore, the structural deterioration and failure of concrete due to rebar corrosion by seawater is not applicable to pervious concrete.

Phosphorus is an essential nutrient for all living organisms as it is a key element of deoxyribonucleic acid (DNA) for growth and reproduction and adenosine triphosphate (ATP) for energy production. On the other hand, nutrient over-enrichment primarily by nitrogen and phosphorus is a non-point source pollution of concern with urban stormwater runoff [29] due to the oxygen depletion and growth of toxic cyanobacteria in water because of the excessive growth and further decomposition of aquatic plants [30]. As such, a need clearly exists to reduce the phosphorus concentration in urban stormwater runoff prior to entering the waterways where it may cause water quality deterioration and other environmental threats.

With the aforementioned needs in mind, the current study produced off-spec FA-added, seawater-mixed pervious concrete (SMPC) and tested it for mechanical (compressive strength), hydrological (permeability), and environmental (phosphorus removal) characteristics. To the best of the authors’ knowledge, this study is the first of its kind for pervious concrete production with the co-utilization of seawater as a mixing water and off-spec FA as a partial cement replacement. The utilization of off-spec FA as a supplementary cementitious material (SCM) and seawater as a mixing water for concrete is expected to improve the sustainability of the concrete industry by substantially saving freshwater usage and managing industrial waste streams.

## 2. Materials and Methods

### 2.1. Main Materials

A Type GU Portland cement was used, and FA was obtained from a coal-fueled power plant (AES, Guayama, PR, USA). The physiochemical characteristics of Portland cements and FA are shown in Table 1. FA can be classified as either Class C or Class F depending on its physiochemical compositions in accordance with the American Society for Testing and Materials (ASTM) C618 [31]. FA containing greater than 70% SiO_2_ + Al_2_O_3_ + Fe_2_O_3_ are classified as Class F, whereas those having a SiO_2_ + Al_2_O_3_ + Fe_2_O_3_ content between 50% and 70% are Class C. In either case, FA needs to contain SO_3_ and loss-on-ignition (LOI) contents lower than 5% and 6%, respectively. It is important to note that FA used in the current study is off-spec FA, not conforming to the ASTM C618 as it contains 45.6% SiO_2_ + Al_2_O_3_ + Fe_2_O_3_, 11.4% SO_3_ and 7.6% LOI. 

Coarse aggregates were limestone gravels in a size of 4.75–9.5 mm. The mass ratio of the coarse aggregates to the binder was fixed at 4:1 to produce pervious concrete. The binder is defined in this study as Portland cement, FA, and nano SiO_2_ (NS). Fine aggregates (e.g., sand) were not used for pervious concrete production in the current study. Seawater was collected and left overnight undisturbed to discard large particulate materials prior to use. Table 2 summarizes the chemical characteristics of seawater. 

BASF MasterGlenium 3030 is a liquid form of water reducer (WR) that was used in the current study. Pervious concrete is generally made of a low water-to-cement ratio of 0.26–0.40 [35] and, therefore, WR is added as an essential component in the production of pervious concrete to create adequate flowability to be quickly discharged from a ready-mixed concrete truck. WR also improves the strength and durability of concrete as it produces a more compact, dense microstructure [36]. Sodium dihydrogen phosphate was purchased from Fisher Scientific and was dissolved in deionized water to make the desired initial concentration at 10 mg/L as $PO_{4}^{-}-P$. 

NS is a white, amorphous powder with a purity of >99% (US Research Nanomaterials, Inc., Houston, TX, USA). It has an average particle size of 20–30 nm, a specific surface area of 180–600 m^2^/g, and a bulk density of <0.10 g/cm^3^. Nanoparticle admixtures, such as NS, are also known to react with CH and increase C-S-H gel formation and to act as a filler by filling the spaces between particles, leading to a denser microstructure, and improving mechanical strength and durability of concrete [37]. Prior to the current study, a preliminary study tested the compressive strength for SMPCs with and without the addition of NS. The results showed that the addition of NS at 2% made an improvement in both the 7- and 28-day compressive strength of SMPCs that contained the off-spec FA at 20% and the WR at 0.2% (Table 3). In this regard, SMPCs in the current study were developed with the addition of NS.

### 2.2. SMPC Mix Optimization

SMPC was prepared by a four-factor, two-level (2^4^) central composite design (CCD) (Table 4) and thereby a total of 30 combinations of independent variable settings were run with 16 factorial points, 8 axial points, and 6 center points. The four factors were the percentages (%wt.) of FA-to-binder, NS-to-FA, seawater-to-binder, liquid-to-binder, and WR-to-binder. In the current study, the binder is defined as the total of the Portland cement and FA and the liquid as the total of seawater and WR. CCD was used to optimize the mix proportioning with the Portland cement, FA, and other admixtures in the production of the pervious concrete specimens [38,39].

A mechanical mixer was used to prepare the pervious concrete specimens. The mixtures in triplicate were cast in a cylindrical plastic mold (10 cm in dia. × 20 cm in height) and the standard rodding consolidation method was used for the compaction of each specimen in accordance with ASTM C192 [40]. The specimens in the mold were placed in an individual airtight plastic bag to minimize moisture loss. After a 24-h curing under an ambient temperature (25 ± 5 °C), the specimens were demolded and further cured in lime-saturated water under an ambient temperature for 7 days prior to the testing of compressive strength and permeability. The compressive strength was tested in accordance with ASTM C39 [41] and the permeability was tested by a constant head method modified from ASTM D2434 [42]. Then, the optimum mix proportion of SMPC was obtained for a maximum-possible 7-day compressive strength and permeability by Response Surface Methodology with the D-optimal desirability functions in Minitab 19. For the experiment of aqueous phosphorus removal, SMPC made of the optimum mix proportion was used. Tap water-mixed ordinary pervious concrete was also produced as the control. The control ordinary pervious concrete (Control PC) was made with the liquid-to-cement ratio of 0.32 and WR-to-cement of 0.35%. Neither FA nor NS were used to produce control pervious concrete. However, it should be noted that the same types of cement, coarse aggregate, and WR were used for both SMPC and Control PC. 

### 2.3. Aqueous Phosphorus Removal

The optimized SMPC was tested for the rate and extent of aqueous phosphorus removal. A kinetic experiment was first conducted to obtain an equilibrium time of phosphorus removal by pervious concrete specimens. Two optimized SMPC specimens were placed in a 5-gallon (19-L) plastic container containing 10 L of a phosphorus solution at 10 mg/L as $PO_{4}^{-}-P$. Aqueous samples were taken at time intervals for 7 days and the phosphorus concentration and pH were measured.

For an isotherm study, 5-gallon plastic containers containing 10 L of a phosphorus solution at 10 mg/L as $PO_{4}^{-}-P$ received the different numbers of the optimized SMPC (1, 2, 3, and 4 specimens). After 72 h of equilibrium time, which was determined from the kinetic study, the aqueous phosphorus concentration and pH were measured in the same manner as in the kinetic study. For both the kinetic and isotherm studies, the Control PC system was run in parallel.

### 2.4. Water Quality Analysis

A Shimadzu Prominence IC system (Kyoto, Japan) was used for the anion analysis ($PO_{4}^{-}-P$ in the phosphorus removal experiment; Cl^−^ and SO_4_^2−^ in seawater). A chromatographic separation was performed at 45 °C with a Shodex SI-52 4E anion column (4.0 mm i.d. × 250 mm) (Showa Denko, Tokyo, Japan). The mobile phase was 3.6 mM sodium carbonate at a flow rate of 0.9 mL/min. The sample injection volume was 20 μL. The sodium concentration in seawater was measured with the sensION Sodium Ion Selective Electrode (HACH, Loveland, CO, USA). The calcium and magnesium concentrations in seawater were determined by HACH Method 8226. The total dissolved solid concentration and pH were measured with the TDSTestr 11 (Oakton Instruments, Vernon Hills, IL, USA) and the Orion 9157BNMD pH probe (Thermo Fisher Scientific, Waltham, MA, USA), respectively. 

### 2.5. Compressive Strength and Permeability of Pervious Concrete

Two response variables (compressive strength and permeability) were tested for pervious concrete specimens. The compressive strength was tested in triplicate in accordance with ASTM C39 (ASTM International, 2016f). Briefly, pervious concrete specimens were placed on a 3000-kN universal testing machine (Forney, Zellenople, PA, USA). The compression load (in lb) at the breakage of the specimen was recorded and the compressive strength was calculated as follows: (1)$$\mathrm{Compressive}\;\mathrm{strength}\;\left(\mathrm{MPa}\right)=\frac{\mathrm{psi}}{145.04}\;$$

The permeability of the pervious concrete specimens was tested in triplicate by a constant head method modified from ASTM D2434 [42]. In a permeameter, the volume of percolated water (V_w_) through the specimen (diameter, D and height, L) was collected for a given time (t), while a constant water head (Δh) was applied to the specimen. Then, permeability (in mm/s) was calculated as follows:(2)$$\mathrm{Permeability}\;\left(\mathrm{mm}/s\right)=\frac{4\cdot V_{w}\cdot L}{\pi \cdot D^{2}\cdot \Delta h\cdot t}$$

Table 5 shows the mix proportions of the optimum SPMC and the Control PC tested in this study.

## 3. Results and Discussion

### 3.1. Optimized SMPC

Table 6 summarizes the results of the compressive strength and permeability for SMPC, which were tested to determine the optimum mix proportion of SMPC. A total of 30 mixtures with different levels of FA/B, NS/FA, L/B, and WR/B were investigated. Note that the 7-day compressive strength of SMPC specimens ranged between 6.9 and 15.1 MPa, falling into a typical compressive strength of ordinary Portland cement pervious concrete (OPC) of 2.8–28 MPa [35]. The permeability of SMPC specimens was measured in a range of 1.98–8.87 mm/s, which was similar to that of OPC (1.4–12.3 mm/s) (ACI, 2010) and fell within the typical range (i.e., 0.5 to 40 mm/s). On average, the SMPC specimens had a 7-day compressive strength at 11.6 MPa and a permeability at 4.2 mm/s. In comparison, the Control PC had a 7-day compressive strength and a permeability at 11.7 ± 0.5 MPa and 6.8 ± 3.2 mm/s, respectively (*n* = 3). Non-significant discrepancies in the compressive strength and permeability were found between the SMPC and the Control PC given the variabilities.

According to the D-optimal desirability functions in Minitab 19 (Figure 1), the optimized SMPC was predicted to have a 7-day compressive strength of 14.0 MPa and a permeability of 5.5 mm/s with the percentages of FA-to-binder at 15.0%, NS-to-FA at 3.0%, liquid-to-binder at 0.338, and WR-to-binder at 0.18%. The validation SMPC specimens were reproduced with the aforementioned optimum mix proportions, and they had a 7-day compressive strength at 13.5 ± 1.3 MPa and a permeability at 6.2 ± 1.4 mm/s (*n* = 3). Therefore, absolute relative percent errors between the prediction and validation were calculated to be 3.7% for the 7-day compressive strength and 11.3% for the permeability with the following equation:(3)$$\left|1-\frac{\mathrm{predicted}\;\mathrm{value}}{\mathrm{validated}\;\mathrm{value}}\right|\times 100\%$$

### 3.2. Long-Term Compressive Strength

Figure 2 compares the compressive strength developments for SMPC, and the Control PC measured at the ages of 7, 14, 28, 56, and 91 days. As shown in the figure, the optimized SMPC had a slightly greater compressive strength at the early ages than the Control PC. However, a long-term maximum compressive strength was measured to be ~16 MPa for both the optimized SMPC and the Control PC. The large standard deviation at 56 days was noted, which is attributed to the small number of samples tested (*n* = 3). A partial replacement of cement with FA typically enables concrete to reduce the rate of strength development and to gain the strength to a greater extent at later ages. This is mainly attributed to the pozzolanic reactivity of FA by which portlandite (Ca(OH)_2_) is reacted with the silica of FA to form additional calcium-silicate-hydrate gel (CSH) at later ages [14,39]. However, the said typical trend of compressive strength development was not observed in the current study with SMPC; rather, the SMPC exhibited slightly higher strengths up to 28 days. This is attributed to the effect of NS used as a filler, which makes concrete less porous. Moreover, accelerated NS–cement hydration could be another reason that led to the high early strength gain via the formation of microstructural C-S–H gel [43]. Further studies are warranted to elucidate mechanisms for an early strength development of SMPC in the presence of off-spec FA and NS. 

### 3.3. Rate of Aqueous P Removal

The rate of aqueous phosphorus (as $PO_{4}^{-}-P$) removal was determined at an initial phosphorus concentration at 10 mg/L as $PO_{4}^{-}-P$ for both SMPC and Control PC. As shown in Figure 3, for both cases, aqueous phosphorus concentrations were exponentially decreased at a first-order removal constant of ~0.03 h^−1^ and reduced by 90% after 72 h of contact time. Vazquez-Rivera et al. [39] also reported a first-order phosphorus removal constant between 0.028 and 0.031 h^−1^ from their kinetic study with pervious concrete containing FA and nano-iron particles. According to Wang et al. [44], precipitation in the form of hydroxyapatite (Ca_10_(PO_4_)_6_(OH)_2_) and amorphous calcium phosphate (Ca_3_(PO_4_)_2_) is responsible for the aqueous phosphorus removal by cement-based materials. The current study also noticed precipitation at the end of the phosphorus removal experiment. Calcium (Ca^2+^) is a key element to form amorphous phosphates. Portlandite (Ca(OH)_2_) is one of the major hydrates formed during cement hydration and is very soluble to water resulting in an increase in the OH^−^ concentration in water.
(4)$$\mathrm{Ca}{\left(\mathrm{OH}\right)}_{2}⇌{\mathrm{Ca}}^{2+}+{\mathrm{OH}}^{-}$$
(5)$$\mathrm{pH}=14+\log\left[{\mathrm{OH}}^{-}\right]$$
where [OH^−^] is the hydroxide concentration in water (mol/L).

There was an increase in pH to ~11.0 from the initial 7.7 after 24 h of contact time during the kinetic experiment in the current study. However, it is premature to account the precipitation in the form of Ca_10_(PO_4_)_6_(OH)_2_ and Ca_3_(PO_4_)_2_ as the phosphorus removal mechanism as the study did not characterize the precipitates. It could have been attributed to other minerals present in seawater. A further investigation is needed to clarify this speculation.

### 3.4. Isotherm of Aqueous P Removal

An isotherm study was conducted at room temperature (20 ± 2 °C) for 72 h of contact time which was determined from the kinetic study. A non-linear removal phenomenon was observed (Figure 4a) for both SMPC and Control PC and therefore their isotherm data were fitted by the following Freundlich equation as shown in Figure 4b.
(6)$$q_{e}=K_{f}\cdot C_{e}^{1/n}$$
where, *q_e_* is the removed phosphorus amount at equilibrium (mg/kg), *K_f_* is the Freundlich constant (mg^1−1/n^ kg^−1^ L^1/n^), *C_e_* is the aqueous phosphorus concentration at equilibrium (mg/L), and 1/n is a dimensionless parameter. The *K_f_* and 1/n for SMPC were found to be 15.72 and 0.4357, respectively and those for the Control PC were 16.63 and 0.4272, respectively. Based on the 1/n’s which were less than 1, it can be said that both SMPC and Control PC favored phosphorus removal. The *K_f_* indicates a removal capacity, but they are only comparable when the 1/n’s are the same because of the dependence of the unit of *K_f_* on 1/n. As the values of 1/n in this study were similar for SMPC and Control PC (i.e., 0.4357 vs. 0.4272), it is construed that SMPC had a slightly reduced phosphorous removal ability than the Control PC, judged by its lower *K_f_* of 15.72, compared to 16.63 of the Control. These findings are in good agreement with Wu et al. [45] who reported the potential of adsorptive concrete aggregates for phosphorus removal. For example, granular aggregates of peach shell had a *K_f_* of 16.6 and a 1/n of 0.719, whereas light weight silica aggregates had a *K_f_* and 1/n of 17.4 and 0.720, respectively (Wu et al., 2021). On the other hand, Vazquez-Rivera et al. [39] reported a *K_f_* of 2.48 and a 1/n of 1.7 from aqueous phosphorus removal with pervious concrete optimized with nano-iron and FA.

## 4. Conclusions

Cement and water are key ingredients to produce cement-based infrastructure. However, the production of cement creates environmental damage in terms of carbon dioxide release and the use of freshwater for mixing cement-based materials is unfavorable considering the shortage of freshwater resources worldwide. FA has been used as a partial replacement of Portland cement in sustainable concrete production. However, those off-spec FAs that are not in compliance with standards such as ASTM are typically landfilled, which can potentially cause different environmental and health problems. Based on the findings, the following conclusions and recommendations can be made:This study demonstrated a cleaner production of pervious concrete with seawater as a mixing water and an off-spec FA as a partial cement replacement to simultaneously attain water, waste, and concrete sustainability;The permeability of SMPC specimens ranged from 1.98 to 8.87 mm/s with the maximum compressive strength of ~16 MPa, which was found to be practically acceptable;The optimized SMPC had a slightly greater compressive strength at the early ages than the Control PC;For both SMPC and Control PC, aqueous phosphorus concentrations were dramatically decreased by 90% after 72 h of contact time, indicating that SMPC showed a plausible potential to improve water quality as a green infrastructure to manage stormwater runoff;Future studies are warranted to elucidate the mechanisms of aqueous P removal by FA-added, seawater-mixed pervious concrete, to test other waters including intermediate or reject seawater from desalination processes, and to understand the effect and fate of the constituents of concern in seawater such as microplastics.

## Figures and Tables

**Figure 1 materials-15-01407-f001:**
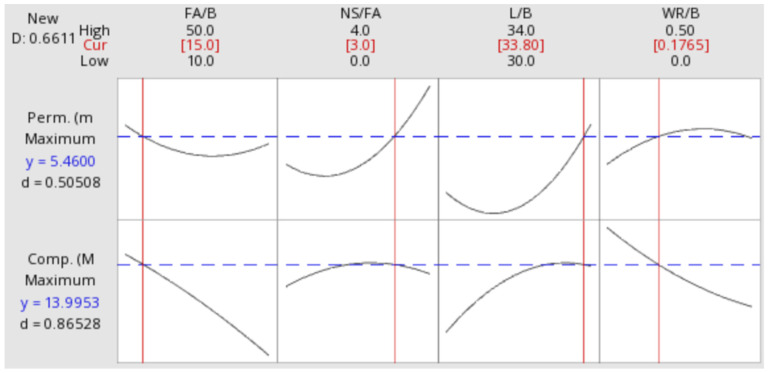
An optimum mix proportion of SMPC for maximum-possible permeability and compressive strength.

**Figure 2 materials-15-01407-f002:**
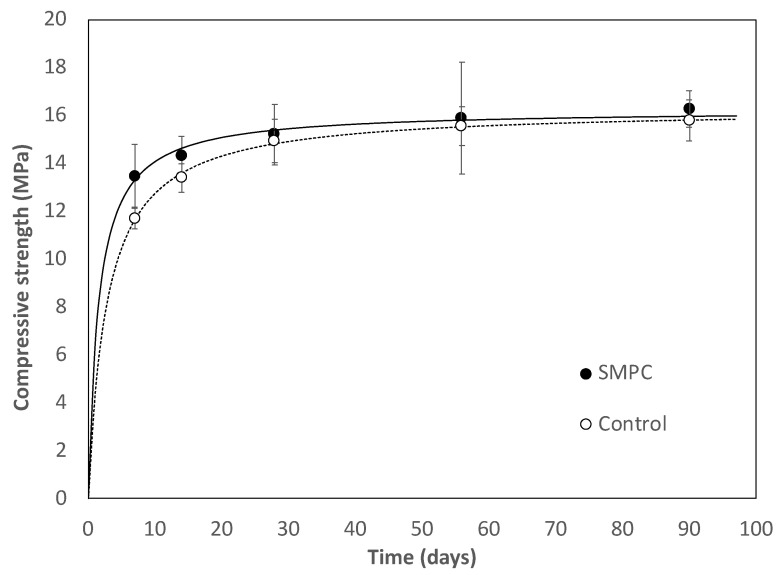
Development of compressive strength.

**Figure 3 materials-15-01407-f003:**
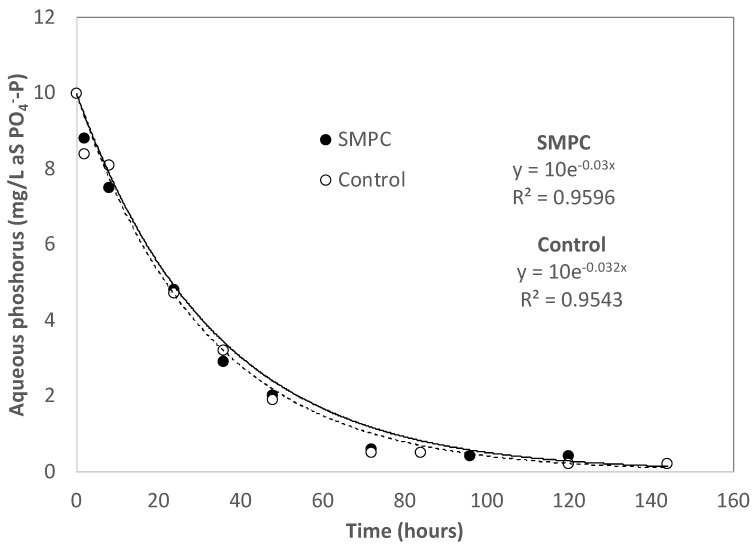
The rate of aqueous phosphorus removal by pervious concrete.

**Figure 4 materials-15-01407-f004:**
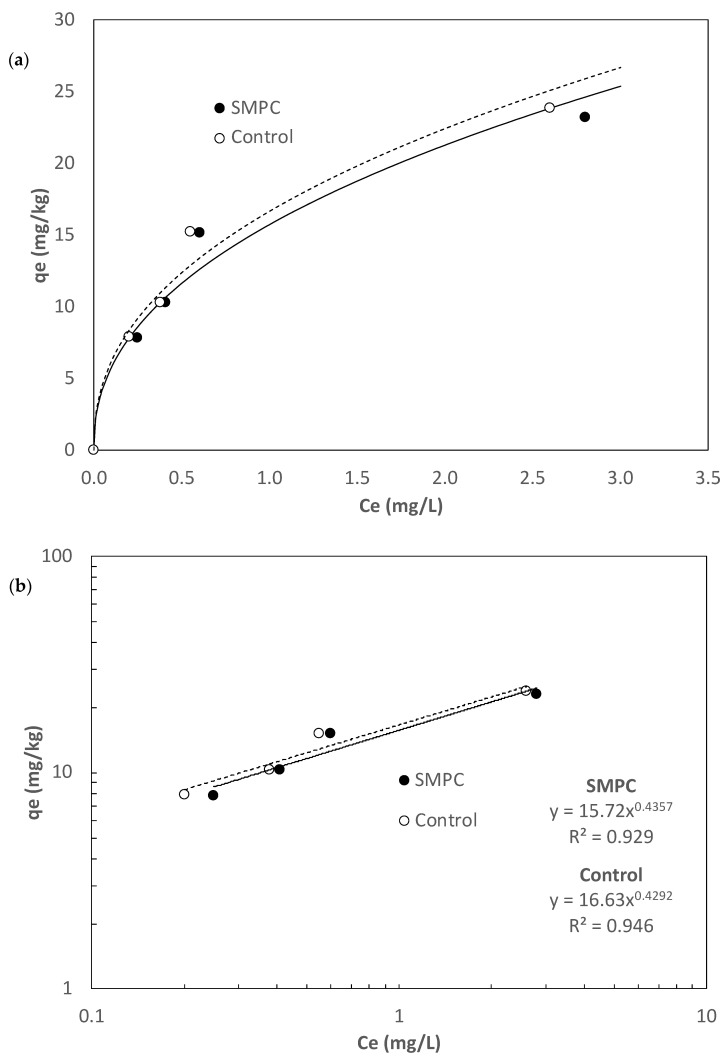
Results from the isotherm study: (**a**) a non-linear isotherm data and (**b**) isotherm data fitted with the Freundlich equation.

**Table 1 materials-15-01407-t001:** Characteristics of Portland cement and FA used in the current study.

Properties	Portland Cement	Off-Spec FA
Mineralogical composition (% wt.)		
SiO_2_	19.8	30.8
Al_2_O_3_	5.1	9.9
Fe_2_O_3_	3.1	5.0
CaO	67.3	39.6
MgO	0.8	0.4
K_2_O	-	1.0
Na_2_O	-	0.9
SO_3_	2.7	11.4
TiO_2_	-	0.5
P_2_O_5_	-	0.1
Loss-on-ignition (% wt.) ^a^	6.8	7.6
Blane (m^2^/kg) ^b^	488	441
Fineness (% wt.) ^c^	92.5	73.7

^a^ Weight loss due to heating at 900–1000 °C (1650–1830 °F) until a constant weight is obtained, according to ASTM C114 [32]. ^b^ A measurement of the surface area, that is referred to as a fineness measure, according to ASTM C204 [33]. ^c^ Wet sieve percentage passing the No. 325 (45 µm) sieve, according to ASTM C430 [34].

**Table 2 materials-15-01407-t002:** Concentrations of ions in seawater ^a^ (g/L).

Cl^−^	SO_4_^2−^	Ca^2+^	Mg^2+^	K^+^	Na^+^
18.6	2.1	4.4	1.1	0.4	4.3

^a^ pH = 7.9, total dissolved solids = 30.5 g/L.

**Table 3 materials-15-01407-t003:** Preliminary results of the compressive strength of SMPCs with and without the addition of NSa. The data shown are the averages with standard deviations (*n* = 3).

	Compressive Strength (MPa)	
	7-Day Curing	28-Day Curing
SMPC with 2% NS	11.5 ± 0.3	13.8 ± 0.5
SMPC without NS	9.7 ± 0.2	12.4 ± 0.3

**Table 4 materials-15-01407-t004:** Four-factor, two-level central composite design for SMPC mix optimization.

Factors	Levels (% wt.)				
	(−) Axial	Low	Center	High	(+) Axial
Fly ash/binder ^a^	10	20	30	40	50
NS/fly ash	0	1	2	3	4
Liquid ^b^/binder	30	31	32	33	34
Water reducer/binder	0	0.13	0.25	0.38	0.50

^a^ binder = Portland cement + fly ash + NS. ^b^ liquid = seawater + water reducer.

**Table 5 materials-15-01407-t005:** The mix proportions ^a^ of the optimum SPMC and Control PC (per m^3^).

	Gravel (kg)	Cement (kg)	FA (kg)	NS (kg)	Seawater (L)	Tapwater (L)	WR (L)
SMPC ^b^	1590.0	336.1	59.6	1.8	133.6	-	0.7
Control PC ^c^	1590.0	397.5	-	-	-	125.8	1.4

^a^ FA: fly ash, NS: nanoSiO2, L = seawater (or tapwater) + WR, WR: water reducer, B: binder = (FA + cement + NS). ^b^ FA/B 15%, NS/FA 3%, L/B 0.338, and WR/B 0.18%. ^c^ L/Cement 32% and WR/Cement 0.35%.

**Table 6 materials-15-01407-t006:** Test results of the compressive strength and permeability for SMPC.

Run #	Factors and Levels (% wt.) ^a^				Responses ^b^	
	FA/B	NS/FA	Liquid/B	WR/B	Compressive Strength (MPa)	Permeability (mm/s)
1	30	2	32	0.25	14.2 ± 1.2	2.36 ± 0.87
2	40	1	33	0.38	11.2 ± 2.0	2.24 ± 1.08
3	20	3	31	0.13	12.1 ± 1.9	2.74 ± 0.90
4	20	3	33	0.38	12.3 ± 2.8	3.96 ± 1.35
5	30	2	32	0.25	11.1 ± 1.1	2.85 ± 0.70
6	40	1	31	0.13	12.1 ± 1.4	5.00 ± 1.66
7	40	3	33	0.13	12.7 ± 4.0	3.52 ± 0.83
8	40	3	31	0.38	12.6 ± 1.1	3.27 ± 0.41
9	20	1	31	0.38	12.8 ± 0.7	2.77 ± 0.81
10	20	1	33	0.13	14.4 ± 1.6	2.69 ± 0.16
11	30	2	32	0.25	10.6 ± 0.4	2.11 ± 0.34
12	40	1	31	0.38	9.60 ± 1.0	5.82 ± 0.39
13	20	1	33	0.38	11.2 ± 2.0	3.13 ± 0.44
14	40	1	33	0.13	11.4 ± 0.5	3.39 ± 1.58
15	20	3	31	0.38	14.1 ± 2.0	2.40 ± 0.24
16	30	2	32	0.25	11.3 ± 1.2	4.24 ± 0.27
17	20	3	33	0.13	15.1 ± 2.2	2.62 ± 0.27
18	40	3	33	0.38	10.8 ± 1.8	3.50 ± 0.90
19	40	3	31	0.13	9.20 ± 2.4	5.82 ± 0.39
20	20	1	31	0.13	12.1 ± 1.9	1.98 ± 0.62
21	30	2	34	0.25	10.8 ± 1.8	5.23 ± 0.83
22	30	2	32	0.00	9.5 ± 1.2	6.59 ± 0.24
23	30	2	30	0.25	8.1 ± 1.1	8.87 ± 1.04
24	10	2	32	0.25	14.1 ± 3.1	3.57 ± 1.61
25	50	2	32	0.25	6.90 ± 1.1	8.70 ± 3.02
26	30	2	32	0.50	13.1 ± 2.8	3.00 ± 0.46
27	30	0	32	0.25	10.7 ± 0.7	6.91 ± 0.25
28	30	2	32	0.25	12.0 ± 1.7	4.40 ± 0.59
29	30	4	32	0.25	9.10 ± 1.0	6.76 ± 1.17
30	30	2	32	0.25	12.8 ± 1.5	5.66 ± 1.08

^a^ FA: fly ash, NS: NS, Liquid = seawater + WR, WR: water reducer, B: binder (FA + cement + NS). ^b^ Response data shown are the average ± standard deviation of triplicate specimens.

## Data Availability

Not applicable.

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
