# Peer review of "Fly Ash-Added, Seawater-Mixed Pervious Concrete: Compressive Strength, Permeability, and Phosphorus Removal"

_materials, 2022, doi:10.3390/ma15041407_

Round 1

Reviewer 1 Report

The article seems very good to me, it has a good experimental and scientific contribution. It has good bibliographic support.

I have the following observations in general:

  1. Increase references regarding the use of seawater in concrete mixtures.
  2. Add photos of the materials or the experimental part –
  3. Mention the equipment used in the experimentation
  4. Review of the units of “seconds”, is it "sec" or "s”?
  5. Review of the edition and format in general
  6. Extend the discussion of results and conclusions

Reviewer 2 Report

This paper deals with pervious concrete made with fly ash and seawater. However, it lacks novelty and must be improved before being considered again.

First, the fly ash content is not high (15%), so it can hardly be called a breakthrough.

Nano Silica content is 3%, which is too high and costly.

The water to binder ratio seldomly uses a percentage, but a fraction.

Introduction:

The key drawback is the references are not update ones, which indicates that the author did not fully pay attention to the recent progress of the topic concerned. Just for example:

The first sentence lacks support, how to prove is it “best”. In addition, better with a reference, for example Pervious concrete: Effect of porosity on permeability and strength.

Second paragraph: Improper disposal of FA disrupts ecosystems, pollutes soil and water, and causes other environmental threats (Ilic et al., 2003; Temini et al., 1995); more recent reference about FA shall updated. See: Carbon fiber reinforced geopolymer (FRG) mix design based on liquid film thickness

Third paragraph: “A water-to-cement ratio typically falls in 0.4 to 0.6. “ Pls add references to show “typically”, for example: articles titled Testing of concrete by rebound method: Leeb versus Schmidt hammers; Fibre factors governing the fresh and hardened properties of steel FRC; Crack mitigation utilizing enhanced bond of rebars in SFRC.

“However, the applicability of seawater in concrete pro- duction is limited due to its high chloride contents that induce corrosion of reinforcing steel bars (Kaushik and Islam, 1995) “ consider replace/add a more recent work, e.g., see: Seawater cement paste: Effects of seawater and roles of water film thickness and superplasticizer dosage.

In the last paragraph of Introduction: pls highlight the innovation points of this research, and contributions to scientific community.

Section 2.2.: Pls give a new Table showing the mix proportions in one cubic meter roughly. In addition, as paste volume is one of the key design parameter. Please calculate and CPV for each mix. The calculation method can be referred to a popular article titled Effect of paste volume on fresh and hardened properties of concrete published in CBM.

Fig. 2: the S.D. is very large. But the average value is very consistent. Pls explain how the results were obtained. (or present individual values before taking average?)

Discussions shall be enriched.

The relationship between strength and permeability shall be emphasized. See: Pervious concrete: Effect of porosity on permeability and strength.

The conclusion is not strong enough and shall highlight the findings better in numerical values.

Reviewer 3 Report

This is an interesting and well elaborated paper focused on the design of pervious concrete for moderation of storm-water in the form of pavements and removal of phosphorous. The presented results and conducted research demonstrated a cleaner production and application of FA-added, seawater-mixed pervious concrete to simultaneously attain water, waste, and concrete sustainability. The paper is of high quality and I believe, it can be accepted for publication in Materials in its present form. The authors made really good job.

Author Response

General Comments:

This is an interesting and well elaborated paper focused on the design of pervious concrete for moderation of storm-water in the form of pavements and removal of phosphorous. The presented results and conducted research demonstrated a cleaner production and application of FA-added, seawater-mixed pervious concrete to simultaneously attain water, waste, and concrete sustainability. The paper is of high quality and I believe, it can be accepted for publication in Materials in its present form. The authors made really good job.

 Response:

The authors would like to express deep appreciation to the reviewer for the time, consideration, and valuable comments.

Reviewer 4 Report

The paper proposed to evaluate the strength and permeability when a proportion of a mix of fly ash and seawater is added into pervious concrete. The main focus was phosphorus removal. It is an unlike study, and tests involving components beyond off-spec represent an early base to developing researches. 

This reviewer would like to point out some issues for understanding and improving the paper.

Initially, when proposing the use of a new material/mixture, the minimum required is an adequate and satisfactory mechanical behaviour. Permeability is an essential requirement for this type of material when it comes to permeable concrete.

For a better understanding of the adopted method, the author must make a flowchart indicating the steps followed and the standards adopted.

One of the proposals was the use of seawater to preserve freshwater. With the various existing technologies to desalinate seawater, why such water was not considered in the study? Seawater has several contaminants, such as microplastics. What could impact the environment when the material is disposed of in landfills or recycling?

Permeability should be described in terms of voids and non-interconnected voids.

The author must show the study's main findings linked with the study contribution.

Only the phosphorus removal tests/results are insufficient to support the research. It is vital to perform lixiviation tests to improve the study.

Reviewer 5 Report

  1. First of all, the title should be re wite for attacking the scientific researchers. The current title does not provide any meaning
  2. The novelty of this manuscript is low and can be improved significantly.
  3. The corrosion study can be improved by referring to and including the following papers:

https://doi.org/10.1016/j.jobe.2021.102281

https://doi.org/10.1016/j.measurement.2021.110318

https://doi.org/10.1016/j.jobe.2020.102029

  1. At which age compressive strength and permeability test was conducted? No information in the manuscript.
  2. How did you measure the pH values? There is no apparatus listed in the manuscript.
  3. The discussion part is not enough and is too short. Can improve
  4. How can you ensure the phosphorous removal from the pH measurement?
  5. The conclusion part can be presented in a point format.

Round 2

Reviewer 2 Report

The author has addressed the comments.

The author can self-check again to maintain zero mistakes.

Author Response

Thanks Reviewer 2 for your time and effort that you have provided to this manuscript review process.

Your comments: The author has addressed the comments. The author can self-check again to maintain zero mistakes.

Responses: We will doublecheck the manuscript to maintain zero mistakes.

Reviewer 4 Report

The authors had efforts to improve the paper. The new manuscript can be published in Materials Journal.

Author Response

Thanks for your time and effort you have provided to the manuscript review process.

Your comments: The authors had efforts to improve the paper. The new manuscript can be published in Materials Journal.

Responses: Thank you again for your excellent service as a reviewer.

Reviewer 5 Report

Accept the manuscript

Author Response

Thanks for your time and effort you have provided to manuscript review process.

Your comments: Accept the manuscript.

Responses: Thanks again for your service as a reviewer.